# Basigin mediation of *Plasmodium falciparum* red blood cell invasion does not require its transmembrane domain or interaction with monocarboxylate transporter 1

Nadine R. King[1], Catarina Martins Freire[1], Jawida Touhami[2], Marc Sitbon[2,3], Ashley M. Toye[1], Timothy J. Satchwell[1]*

**1** School of Biochemistry, University of Bristol, Bristol, United Kingdom, **2** Institut de Génétique Moléculaire de Montpellier, Université de Montpellier, Centre National de la Recherche Scientifique (CNRS), Montpellier, France, **3** Laboratory of Excellence GR-Ex, Paris, France

* t.satchwell@bristol.ac.uk

## Abstract

*Plasmodium falciparum* invasion of the red blood cell is reliant upon the essential interaction of PfRh5 with the host receptor protein basigin. Basigin exists as part of one or more multi-protein complexes, most notably through interaction with the monocarboxylate transporter MCT1. However, the potential requirement for basigin association with MCT1 and the wider role of basigin host membrane context and lateral protein associations during merozoite invasion has not been established. Using genetically manipulated *in vitro* derived reticulocytes, we demonstrate the ability to uncouple basigin ectodomain presentation from its transmembrane domain-mediated interactions, including with MCT1. Merozoite invasion of reticulocytes is unaffected by disruption of basigin-MCT1 interaction and by removal or replacement of the basigin transmembrane helix. Therefore, presentation of the basigin ectodomain at the red blood cell surface, independent of its native association with MCT1 or other interactions mediated by the transmembrane domain, is sufficient to facilitate merozoite invasion.

## Author summary

Malaria is caused by invasion of red blood cells by *Plasmodium* parasites. Invasion by *Plasmodium falciparum* is reliant on an interaction between parasite PfRh5 and host cell basigin. The reasons why basigin specifically is targeted and exploited by the parasite for such an essential process and the importance of its native properties, interactions and context within the red blood cell membrane for mediating invasion are unclear. Through invasion of lab grown reticulocytes in which the natural features and interactions of basigin are disrupted we show that neither the established interaction of this protein with the monocarboxylate transporter MCT1, nor the portion of this protein that resides within or beneath the plasma membrane itself are required for facilitation of successful invasion by the basigin extracellular domain. Presentation of the extracellular domain of basigin alone at the

**Funding:** This study was supported through funding provided by the Medical Research Council (MR/V010506/1) to AMT and TJS and the European Union ITN 'EVIDENCE' grant agreement ID 860436 and in part by funding from Fondation pour la Recherche Médicale (FRM) grant DBI201312285579 (to MS) and National Institutes of Health (NIH) grant DK32094 (to MS). The funders played no role in the study design, data collection and analysis, decision to publish or preparation of the manuscript.

**Competing interests:** MS is inventor on patents describing the use of RBD ligands and a co-founder and head of the scientific board of METAFORA-Biosystems. AMT is a co-founder, director and consultant to Scarlet Therapeutics Ltd, TJS is a co-founder and scientific consultant to Scarlet Therapeutics Ltd.

red blood cell surface, including within different host cell membrane contexts is therefore sufficient for this proteins function in facilitating successful invasion by malaria parasites.

## Introduction

Malaria, a disease caused by *Plasmodium* parasites imposes an enormous global economic and health burden with significant morbidity and mortality, particularly in tropical and subtropical regions. Every year it is responsible for more than 200 million clinical cases accounting for over half a million deaths [1].

Central to the pathogenesis of this disease is the intricate process of red blood cell invasion by *Plasmodium* merozoites, the invasive blood stage form of the parasite. Successful invasion is preceded by a series of specific molecular interactions between proteins on the surface of the parasite and host cell [2]. Chief amongst these in respect of invasion by *Plasmodium falciparum*, the species responsible for the vast majority of fatalities, is basigin, host receptor for the merozoite surface protein PfRh5 [3]. Interaction between these proteins sets the stage for a cascade of events that lead to successful invasion including the release of rhoptry organelles and the establishment of a parasitophorous vacuole. Yet, whilst the strain transcendent essentiality of PfRh5-basigin binding for successful invasion is now well established, the precise role of this interaction in enabling invasion remains controversial and the bases for the selection of basigin as a key actor for such an essential process are unclear.

Basigin, a transmembrane glycoprotein present at around 3000 copies at the red blood cell surface, [4,5] exists not in isolation within the membrane but as part of one or more multiprotein complexes, the most established of which is with the monocarboxylate transporter MCT1 [6–8]. Basigin interacts tightly with MCT1 via its single transmembrane helix, facilitating the plasma membrane delivery, stability and function of this lactate and other monocarboxylates transporting multipass membrane protein [6,7,9]. In one recent report, basigin was found to exist in one of two macromolecular complexes, with MCT1 or the $Ca^{2+}$ ATPase PMCA1/4, with PfRh5 demonstrated to bind to either of these complexes with higher affinity than to isolated basigin ectodomain alone [8]. This suggested a possible role for the lateral interactions of basigin in its native heteromeric complex within the red cell membrane during invasion. However, the potential requirement for basigin association with MCT1 for successful merozoite invasion and the wider role of basigin host membrane context and lateral protein associations in this process has not been established.

In this study we exploit genetic manipulation of primary haematopoietic stem cells and the immortalised erythroblast cell line BEL-A to generate novel host cell reticulocyte phenotypes in which the basigin extracellular domain is presented in different host membrane contexts. Using malaria parasite invasion assays performed on these novel cellular models, we dissect the prospective importance and role of basigin host membrane context, transmembrane helix and its interactions with the monocarboxylate transporter MCT1 within the red cell membrane for successful *P. falciparum* invasion.

## Materials and methods

### Ethics statement

All human blood source material was procured from NHS Blood and Transplant (NHSBT, Filton, Bristol and provided with written informed consent for research use given in accordance

with the Declaration of Helsinki. Ethics approval was granted by National Health Service Health Research Authority, Bristol Research Ethics Committee reference 12/SW/0199.

## Antibodies

Mouse monoclonal antibodies used were as follows: BRIC4 (GPC), BRIC216 (CD55), BRIC222 (CD44), BRIC235 (CD44), KZ1 (CD44), BRIC71 (band 3), BRIC256 (GPA) (all IBGRL hybridoma supernatants used 1:2), HIM6 (basigin) (Biolegend [1:50 flow cytometry), IgG1 control MG1-45 (1:50 Biolegend). Secondary antibodies were allophycocyanin (APC)-conjugated monoclonal rat anti-mouse IgG1 RMG1–1 (Biolegend 1:50) or goat anti mouse IgG poly4053 (Biolegend 1:50). For exofacial detection of MCT1, an RBD - HC2.RBD.rFC (Metafora – 1:5) was used in conjunction with Alexa 647 donkey anti rabbit Poly4064 (1:50 Biolegend).

## BEL-A cell culture

BEL-A (Bristol Erythroid Line–Adult) cells were expanded as previously described [10–12]. In the expansion phase, cells were cultured at a density of $1–3 \times 10^5$ cells/ml in expansion medium, which consisted of StemSpan SFEM (Stem Cell Technologies) supplemented with 50 ng/ml SCF, 3 U/ml erythropoietin, 1 µM dexamethasone (Sigma) and 1 µg/ml doxycycline (Sigma). Complete medium changes were performed every 48 h. Differentiation was conducted as described with modifications. In the differentiation phase, cells were seeded at $1.5 \times 10^5$/ml in differentiation medium (Iscove's Modified Dulbecco's Medium (IMDM; FG0465 Sigma), supplemented with 3 U/ml erythropoietin (Bristol Royal Infirmary), 3 U/ml heparin (Sigma), 0.2 mg/ml holotransferrin (Sigma), 5% v/v Octaplas (Octapharma), 10 µg/ml insulin (Sigma), 100 U/ml penicillin (Sigma) and 100 µg/ml streptomycin (Sigma). For days 0-2 differentiation medium was supplemented with 1 ng/ml IL-3, 10 ng/ml SCF and 1 µg/ml doxycycline). After 2 days, cells were reseeded at $3.5 \times 10^5$/ml in differentiation medium with 10ng/ml SCF and 1 µg/ml doxycycline. On differentiation day 4, cells were reseeded at $5 \times 10^5$/ml in fresh differentiation medium without doxycycline. On differentiation day 7, a complete media change was performed, and cells were reseeded at $1 \times 10^6$/ml in differentiation medium without SCF or doxycycline. On day 9, a complete media change was performed. Cells were leuko-filtered for purification of reticulocytes on Day 10 as previously described [10] and stored in media at 37˚C for up to 72 hours post filtration before use for invasion assays

## CD34+ cell culture

For isolation of peripheral blood mononuclear cells (PBMCs), blood from waste apheresis material was mixed with 0.6% v/v citrate-dextrose solution (ACD; Sigma), diluted 1:1 with Hanks balanced salt solution (HBSS; Sigma) with 0.6% v/v ACD and layered on top of 25 ml PBMC Spin (Sigma). The sample was then centrifuged at 400G, at room temperature (RT) for 35 min. The interface layer consisting of density-purified mononuclear cells was then collected, washed three times in HBSS and resuspended in 12 ml cold Red Cell Lysis Buffer ($NH_4Cl$, 4.15 g/L; EDTA, 0.02 g/L; $KHCO_3$, 0.5 g/L) at 4 ˚C for 10 min, to lyse remaining erythrocytes. Cells were washed twice in PBS and live cells were counted on a CellDrop (DeNovix) using the Trypan Blue dye (Sigma) exclusion test.

CD34+ magnetic cell isolation was performed on the PBMCs according to the manufacturer's protocol for the Direct CD34+ progenitor cell isolation kit (Miltenyi Biotec), to enrich for haematopoietic progenitor cells. Cells were cultured according to a protocol initially described by Griffiths et al [13]. Isolated cells were counted and plated at a density of $1 \times 10^5$ cells/ml in a primary expansion medium. This primary base medium was IMDM (FG0465 Sigma)

supplemented with 3U/ml erythropoietin (Bristol Royal Infirmary), 3U/ml heparin (Sigma), 0.2 mg/ml holotransferrin (Sigma), 3% v/v heat-inactivated Human Male AB Serum (Sigma), 2 mg/ml Human Serum Albumin (HSA; Irvine Scientific), 10 μg/ml insulin (Sigma), 100 U/ml penicillin (Sigma) and 100 μg/ml streptomycin (Sigma), with extra supplementation of 40 ng/ml Stem Cell Factor (SCF; Miltenyi Biotec) and 1 ng/ml IL-3 (R&D Systems) to induce cell proliferation. The cells were incubated at 37 ˚C in 5% $CO_2$ in this primary medium with daily media addition from Day 3 to Day 7 of culture. From Day 8 to Day 12, secondary medium was added instead, which consisted of the same IMDM base supplemented with 40 ng/ml SCF. After Day 13, tertiary medium consisting of the IMDM base without growth factor additions was used to induce terminal erythroid differentiation. On Day 21, reticulocytes were purified through leukofiltration of the culture to remove nuclei and nucleated cells as previously described [13].

## Lentiviral transduction

HEK293T cells (Clontech) were cultured in Dulbecco's Modified Eagle Medium (DMEM) (Gibco) containing 10% fetal calf serum (Gibco). Cells were seeded in 10 cm dishes and transfected using the calcium phosphate precipitation method to prepare lentiviral particles by co-expression of lentiviral packaging vectors pMD2 (5 μg) and pPAX (15 μg) and the lentiviral vector of interest (20 μg). After 24 h, DMEM was removed and replaced with 5 ml fresh media. Virus was harvested after 48 h, concentrated using Lenti-X concentrator (Clontech) according to the manufacturer's protocol and stored at − 80 ˚C. Concentrated virus equivalent to that harvested from half a 10 cm dish of HEK293T cells was added to $2 \times 10^5$ BEL-A cells in the presence of 8 μg/mL polybrene (Sigma) for 24 h. Cells were subsequently washed three times in PBS and resuspended in fresh media. For transduction of primary CD34[+] cells on Day 6, virus equivalent to one 10cm dish per $0.5 \times 10^6$ erythroblasts was used.

## Nucleofection

Knockout of basigin in primary CD34[+] cells was achieved by nucleofection of Cas9, gRNA ribonucleoprotein complexes on Day 2 of expansion culture. RNPs were prepared by addition of 62.5pmol of each of 2 chemically modified sgRNAs (Synthego) targeting basigin, or 125pmol of non- targeting (NT) control guide to 50pmol Cas9 and incubating at 25˚C for 15 minutes. Guide sequences were as follows BSG sgRNA1 UUCACUACCGUAGAAGACCU, BSG sgRNA2 GGCGCUGUCAUUCAAGGAGC or negative control scrambled sgRNA#1 (Synthego). On Day 2 RNP complexes were added to $1 \times 10^6$ PBS washed cells per well of a 16 well nucleofection cassette in a 20ul total reaction volume of nucleofection buffer P3. Cells were nucleofected using the EO-100 program of the 4D Amaxa Nucleofector (Lonza) and returned to expansion media with multiple wells pooled where required. Generation of MCT1 KO BEL-A cell line was achieved by nucleofection with RNPs with sgRNA sequences as follows: ACGUGACUGGCUAGCUGCGU and GCGCAGGCUGGAGUUCCACG.

## Flow cytometry

For flow cytometric assessment of membrane protein expression on BEL-A or CD34[+] derived reticulocytes, mixed differentiated cultures were stained with 5 μg/ml Hoechst 33342 then fixed in 1% paraformaldehyde, 0.0075% glutaraldehyde to reduce antibody binding-induced agglutination before labelling with antibodies as described. For cell surface MCT1 detection, labelling with HC2.RBD.rFC, was performed at 37˚C for 30 mins on live cells, followed by labelling with Alexa 647 conjugated donkey anti rabbit (Poly4064 Biolegend) for 30 minutes at room temperature. Reticulocytes were identified by gating upon Hoechst-negative population.

## FACS

Transduced BEL-A cells expressing mutants of interest were sorted using anti-basigin HIM6 antibody to match surface presentation of basigin ectodomain in expanding BEL-A cells to endogenous levels when possible. For sorting of basigin KO primary erythroblasts, nucleo-fected Day 5 erythroblasts were labelled with HIM6 and APC conjugated secondary antibody and cells sorted based on Draq 7 negativity and absence/low expression of basigin on a BD Influx or Aria II Cell Sorter. Verification of isolation of BSG KO population following turnover of residual protein was conducted following additional days in culture or post enucleation in non-rescued BSG KO control cells.

## Parasite culturing

*P. falciparum* strain 3D7 parasites (MRA-102 BEI Resources, contributed by Daniel J Carucci) were maintained in human erythrocytes at 5% hematocrit using standard culture conditions [14]. The culture medium consisted of RPMI 1640 containing 5.96 g/L HEPES, 2 g/L sodium bicarbonate, and 0.0053 g/L Phenol Red (Sigma), supplemented with 0.05 g/L hypoxanthine (Sigma), 0.025 g/L gentamycin (Sigma), 0.3 g/L L-glutamine (Sigma), and 5 g/L AlbuMAX II (Thermo Fisher Scientific). Cultures were incubated at 37°C in a gas mixture of 5% $O_2$, 5% $CO_2$, 90% $N_2$.

## Reticulocyte invasion assays

Schizont stage parasites were magnetically purified using the Magnetic Cell Separation (MACS) system (Miltenyi Biotec) and added to wells of a round bottomed 96-well plate containing erythrocytes or leukofiltered CD34[+] or BEL-A-derived reticulocytes in culture medium as previously described [10]. Each well contained $1 \times 10^6$ CD34[+] derived reticulocytes, or for BEL-A derived reticulocytes, typically $5 \times 10^5$ cells. Parasitemias ranging from 1-8% were used. Heparin (100 mU/μl final) was used to inhibit invasion in negative controls. After ~ 18 h, invasion was quantified using flow cytometry. For flow cytometry, cells were stained with SYBR Green (1:2000 in culture media; Sigma-Aldrich) for 30 min at 37°C in the dark. Cells were centrifuged, SYBR Green-containing media removed and then fixed for 15 minutes at room temperature as described above before labelling with anti-basigin HIM6 and APC conjugated secondary antibody and acquisition on a MacsQuant 10 VYB cytometer. Invasion was quantified within basigin negative or positive population (for transduced CD34[+] cell derived reticulocytes) as appropriate, based on SYBR green positivity, with heparin control used to correct for background events within the invasion gate for each individual cell type and >100,000 total events acquired per experiment.

## Results

Previous work from our laboratory demonstrated the ability to generate viable *in vitro*-derived reticulocytes with complete absence of basigin expression [10]. Basigin-deficient reticulocytes exhibited no reduction in expression of other key host surface proteins known to play a role in malaria parasite invasion but were shown to be completely refractory to invasion by *Plasmodium falciparum*, thus validating the substantial evidence base regarding the essentiality of this host receptor for successful merozoite invasion.

Basigin exists within many cell types, including human erythrocytes, as part of a hetero-meric complex with the monocarboxylate transporter MCT1 [6–8]. To assess the impact of basigin KO on expression of MCT1, basigin KO BEL-A cells were differentiated to reticulocytes and MCT1 surface expression assessed using a novel receptor-binding domain (RBD)

derived from the HC2/HERV-T human endogenous retrovirus envelope protein sequence (GenBank access AB266802), which specifically binds exofacial determinants of MCT1 (METAFORA-biosystems). HERV-T envelope has been previously shown to use SLC16A1/MCT1 as receptor [15]. Flow cytometry data (Fig 1A) demonstrates that in the absence of basigin, reticulocyte surface expression of MCT1 is reduced by more than 80%. Thus, expression within a cellular context of these two proteins is tightly coupled, raising questions as to the contribution of the loss of MCT1 and/or the MCT1-basigin interaction for abrogated invasion in the absence of basigin. To assess the presence of basigin within the membrane in the absence of MCT1, BEL-A cells were nucleofected with Cas9:gRNA RNPs targeting the *SLC16A1/MCT1* gene, allowing for the generation of MCT1 KO reticulocytes (Fig 1B). These reticulocytes however were found to exhibit a 75% decrease in surface expression of BSG, illustrating co-dependence of these proteins on one another for expression. Since shRNA-mediated knockdown of BSG by just 50% reduces invasive susceptibility of reticulocytes by 80% [3], invasion studies of these cells would not be informative about the putative involvement of MCT1 or its interaction with basigin in invasion.

Therefore, in order to explore the importance of the native membrane context of basigin for successful invasion and to uncouple presentation of the basigin extracellular domain (binding site for PfRh5) from the native interaction of this protein with MCT1, a more elaborate strategy was required. Hybrid versions of basigin were designed; specifically, two constructs in which the transmembrane (TM) helix that mediates interaction with MCT1, and the cytoplasmic domain of basigin were I) substituted with the TM and cytoplasmic tail of another erythroid membrane protein – Glycophorin C (BSG-GPC), or II) removed entirely by replacement with a GPI anchor sequence (BSG-GPI) (Fig 2C). The rationale for these designs lies in the fact that neither should be capable of binding MCT1, yet each alters the native membrane context of the exposed basigin ectodomain differently. To validate the ability of these constructs to present at the cell surface, lentiviral transduction of the previously established clonal BSG KO BEL-A cell line was performed, cell surface presentation of the basigin extracellular domain of the BSG-GPC and BSG-GPI rescues was confirmed and FACS used to select a population with BSG surface expression matched to that of unedited expanding BEL-A controls. BSG KO cells lentivirally rescued with WT basigin (WT-BSG) served as a control.

BSG-GPI and BSG-GPC hybrid proteins were successfully trafficked to the plasma membrane (Fig 1D). The presence of Hoechst-negative enucleated reticulocytes within the differentiated population demonstrated that BEL-A cells expressing these hybrids in the absence of endogenous basigin can undergo terminal erythroid differentiation and generate reticulocytes that present the basigin extracellular domain at close to normal levels (Figs 1D and S1 Fig). Of note, expression of basigin on reticulocytes, which are larger than erythrocytes, is higher than that on erythrocytes [16] and as such is not limiting for invasion in any of the reticulocyte subsets generated despite minor variations. Assessment of MCT1 expression in the corresponding populations (Fig 1E and 1F) demonstrates that the MCT1 loss accompanying BSG KO is only rescued by reintroduction of WT-BSG, as BSG-GPI and BSG-GPC hybrids both lack the TM domain crucial for this interaction. Thus, we demonstrate the ability to generate novel host cellular invasion models in which presentation of the PfRh5 binding site is uniquely uncoupled from the native interaction of BSG with MCT1. To assess the effect of expression of these hybrids has on expression and presentation of other host membrane proteins of interest, labelling with a panel of antibodies was performed and expression quantified by flow cytometry. Fig 1G demonstrates no significant alterations in expression of band 3, GPA, GPC in reticulocytes rescued with BSG-GPI compared to unedited or WT rescue cells. Of these proteins, only GPC appears reduced in reticulocytes expressing BSG-GPC (suggestive of competition for membrane incorporation between endogenous GPC and the hybrid possessing its TM helix

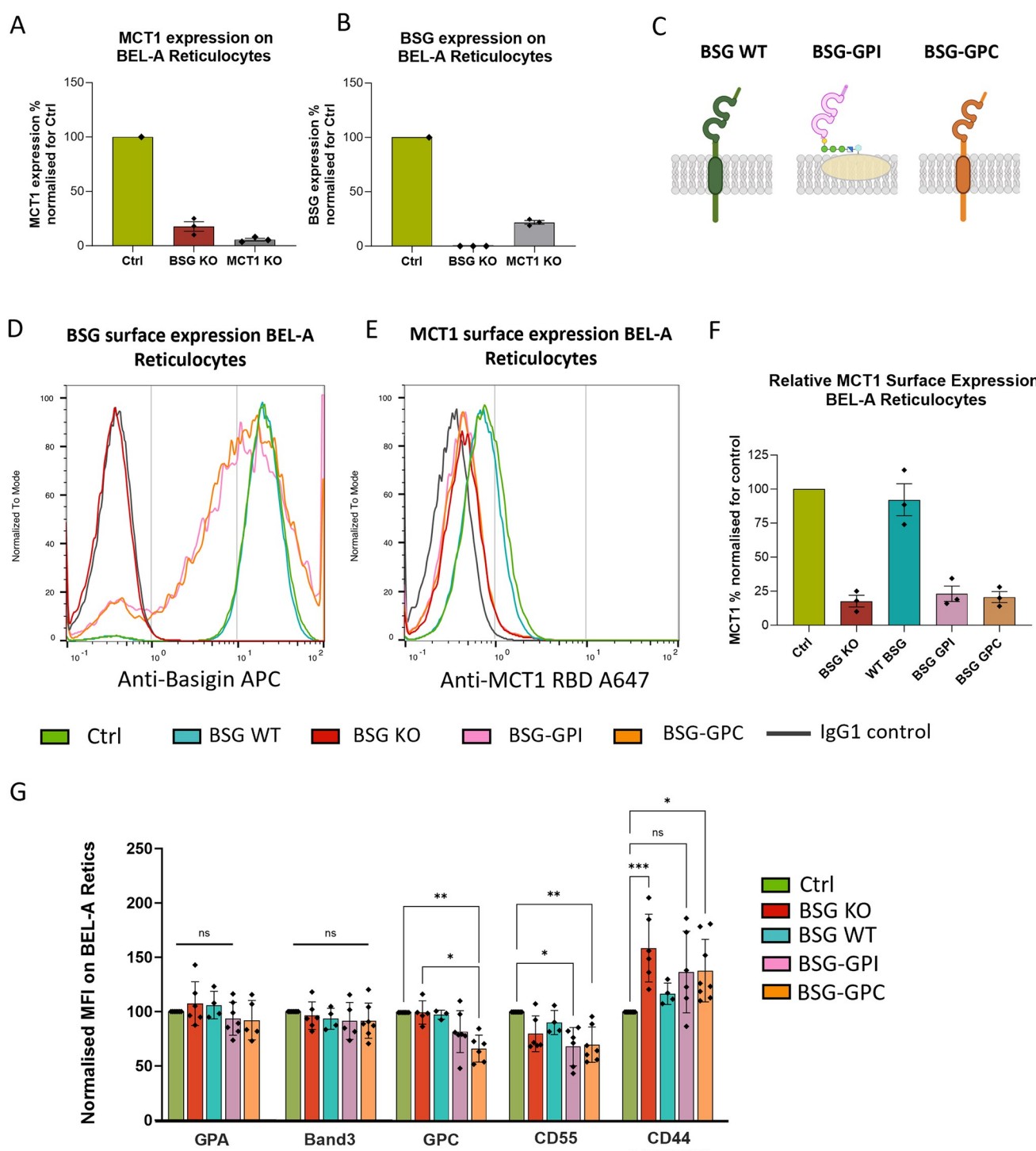

**Fig 1. Presentation of the basigin extracellular domain in reticulocytes can be uncoupled from the native interaction and dependence on MCT1.** A) MCT1 surface expression on unedited, BSG KO and MCT1 KO BEL-A derived reticulocytes assessed by flow cytometry using HC2.RBD.rFC labelling **B)** Basigin surface expression on unedited, BSG KO and MCT1 KO BEL-A derived reticulocytes assessed with HIM6 antibody **C)** Pictoral representations of designed BSG-hybrid proteins. Figure created using BioRender.com **D)** Representative flow cytometry histograms for basigin surface expression on indicated Hoechst negative reticulocyte populations **E)** Representative flow cytometry histograms for MCT1 surface expression on indicated Hoechst negative reticulocyte populations **F)** Normalised relative expression levels of MCT1 (Median fluorescent Intensity MFI normalised for each sample normalised to unedited control reticulocytes) for indicated mutants. **G)** Bar chart illustrating expression of red blood cell membrane proteins on indicated BEL-A derived reticulocytes normalized to unedited cells. Data from at least 3 independently differentiated experiments are presented. A Kruskal-Wallis

comparison followed by Dunn's multiple comparison correction was performed to test for differences between groups. *p* < 0.05 was considered statistically significant.

and cytoplasmic tail confirming altered membrane context compared to WT-BSG). CD55 is mildly reduced in BSG-GPI and GPC expressing cells whilst CD44 presentation as assessed with the BRIC222 antibody, (but not KZ1 or BRIC235 – S2 Fig) appears to be variably increased in BSG KO reticulocytes, restored to endogenous presentation by WT BSG rescue but not by BSG-GPI or GPC, further indicative of disrupted native context of the hybrid proteins.

Having established the viability of reticulocytes with basigin uncoupled from MCT1, we sought to assess the susceptibility of these cells to invasion by *Plasmodium falciparum*. Generation of modified *in vitro* derived reticulocytes can be achieved through differentiation of both immortalised erythroblast cell lines such as BEL-A or of primary CD34$^+$ haematopoietic stem cells. Each system has its own respective advantages and disadvantages [17]. Efficiency of differentiation and high rates of enucleation of primary cells currently facilitate a much greater cost effective yield of leukofiltered reticulocytes, providing more material for experimentation, although this is countered by variable transduction efficiency and reduced control over rescue overexpression levels. Contrastingly BEL-A cells are readily transducible, can be FACS sorted

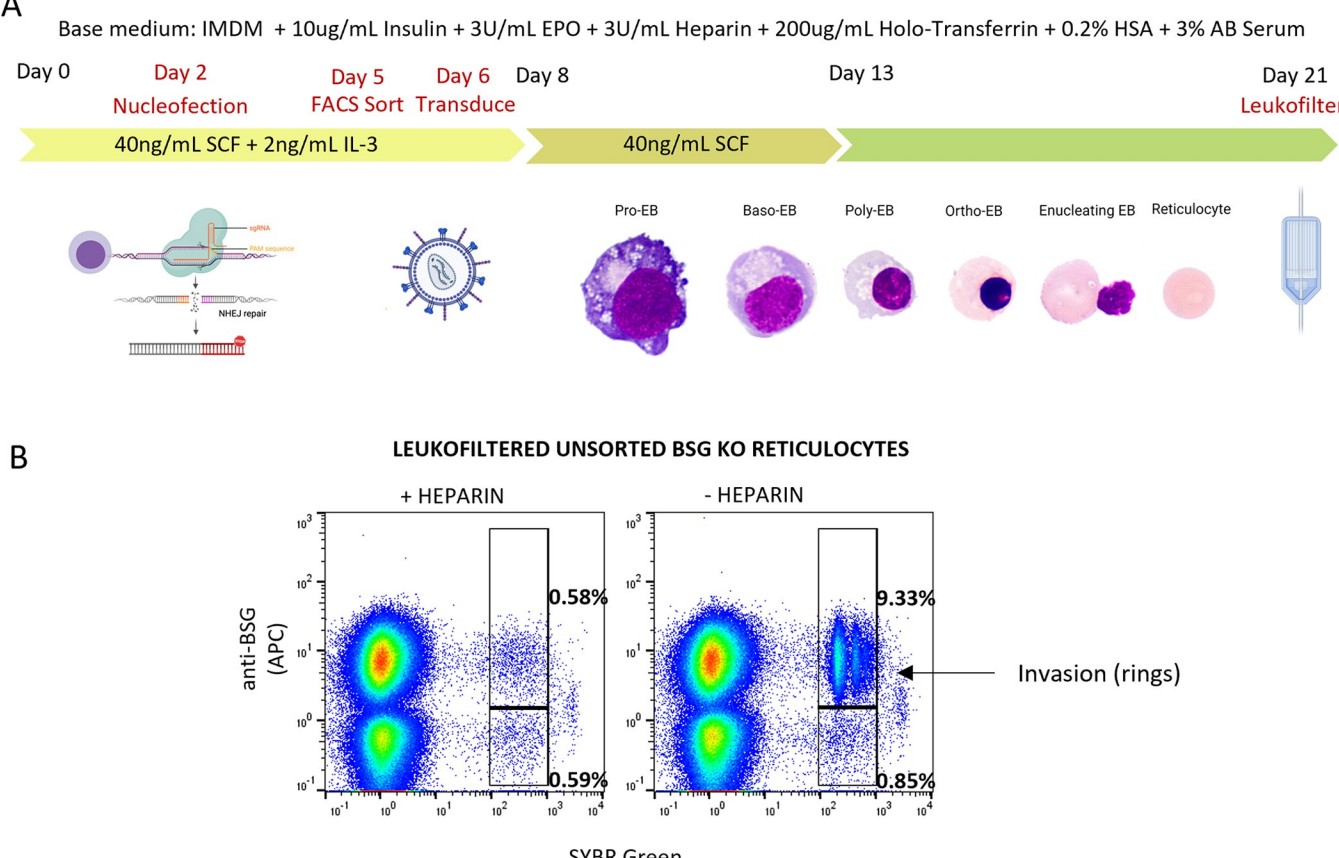

**Fig 2. Establishment of protocol for combined knockout and rescue experiments to assess invasive susceptibility in reticulocytes derived from modified primary erythroblasts. A)** Schematic of experimental procedure for derivation and invasion studies of novel hybrid BSG mutant reticulocytes. Figure created using BioRender.com **B)** Flow cytometric gating strategy for quantification of invasion in BSG KO and rescue CD34$^+$ cell derived reticulocyte populations.

to isolate populations entirely expressing a construct of interest at uniform levels and cryopreserved for repeated differentiation yet exhibit disrupted terminal differentiation with reduced enucleation efficiency compared to primary cells, limiting the yield of purifiable reticulocytes that can be obtained, particularly where cells may have been heavily manipulated.

We began by exploiting the primary cell system. CD34[+] cells 2 days post isolation were nucleofected with CRISPR Cas9:gRNA ribonucleoprotein complexes in order to knockout endogenous basigin. On Day 5, cells were sorted using FACS to isolate the knockout population and transduced on Day 6 with lentiviral particles for exogenous rescue with WT-BSG, BSG-GPI or BSG-GPC constructs. A proportion of cells were untransduced and served to enable validation that the basal population progresses to complete BSG KO reticulocytes that are refractory to invasion. Mixed erythroblast populations (comprising successfully transduced and untransduced cells) are expanded and differentiated according to previously established protocols and leukofiltered to remove nucleated cells and pyrenocytes on Day 21. Fig 2A provides a summary of the experimental procedure. Malaria parasite invasion assays were set up in 96 well plates by addition of MACS magnet purified 3D7 schizonts to 1x10$^6$ total reticulocytes per assay at a variety of starting parasitemias in triplicate. For each condition a heparin treated negative control was included to enable correction for spurious background events within the invasion gate upon assessment of invasion by flow cytometry. Assays post invasion were stained with SYBR green, fixed and labelled with anti-basigin antibody HIM6 and APC conjugated secondary antibody. Antibody labelling allows for discrimination of BSG KO reticulocytes from those expressing BSG constructs of interest. Successfully invaded reticulocytes within the respective populations can then be assessed within the same well based upon SYBR Green positivity. Fig 2B provides an illustrative example of this strategy from an experiment conducted with no FACS purification of BSG KO. Note robust levels of invasion in cells positive for BSG that is absent within the KO population, the accompanying heparin control enables identification and subtraction of background within the invasion gate. Application of this strategy allows for robust assessment of invasion even in the context of variable and/or low lentiviral transduction efficiencies inherent to work with primary erythroid cells. Representative flow cytometry dot plots and histograms illustrating populations of reticulocytes with BSG construct rescue and surface expression equal to or greater than that of unedited reticulocytes and red blood cells are shown in S3 Fig.

Fig 3A illustrates that using assays set at a range of parasitemias invasion is absent or negligible within the BSG KO gated population yet robustly detected within populations in which the basigin extracellular domain is presented at the cell surface. Final parasitemias increase but do not plateau with increasing initial schizont levels excluding saturation effects that may mask impaired invasion phenotypes. Pooled invasion data from multiple experiments normalised to levels in reticulocytes derived from non-targeting control gRNA nucleofected CD34[+] cells demonstrates robust and similar levels of invasion in BSG KO cells rescued with WT-BSG, BSG-GPI and BSG-GPC (Fig 3B). These data demonstrate that the transmembrane domain of basigin and by extension its interaction with MCT1, PMCA1/4 and other proteins that may interact via this domain is not required for successful mediation of merozoite invasion by *P. falciparum*, with presentation of the extracellular domain alone within the membrane sufficient for invasion.

To provide confirmation of this striking result in another cell system, BEL-A cell lines expressing the WT, BSG-GPI and BSG-GPC constructs with 100% positivity were differentiated alongside BSG KO and unedited BEL-A controls. Despite low levels of enucleation within these lines we were able to scale up cultures to isolate leukofiltered BEL-A derived reticulocytes from each of the lines for limited invasion assays. Fig 4A summarises data obtained by flow cytometry that recapitulates the data shown in Fig 3 with robust and similar levels of invasion

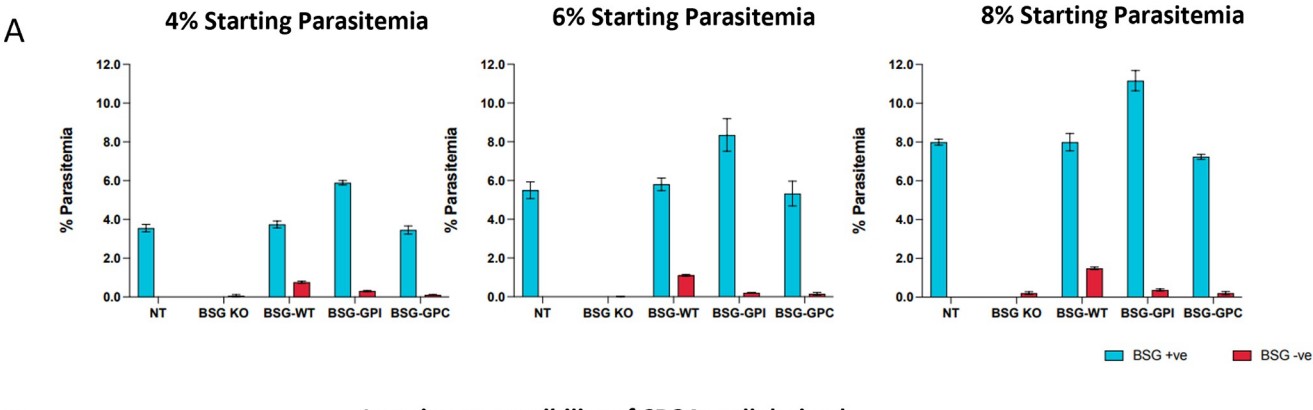

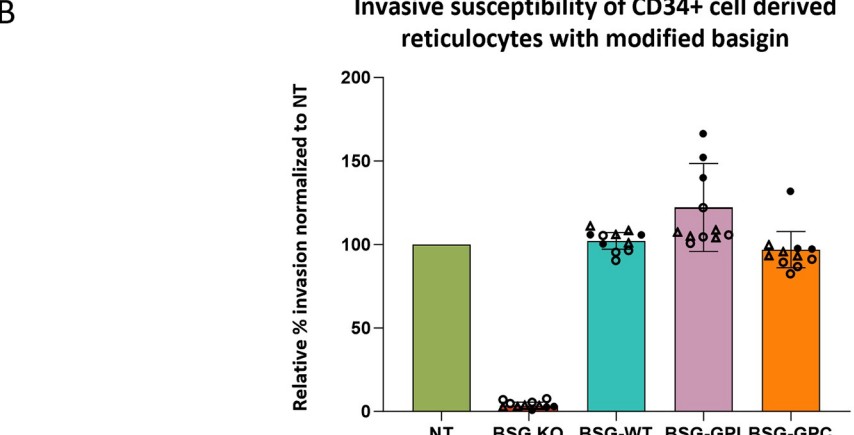

**Fig 3. The basigin transmembrane domain is dispensable for facilitation of merozoite invasion. A)** Parasitemia quantified by flow cytometric evaluation based on SYBR Green positivity within BSG KO and rescued populations as indicated at different starting parasitemias. Parasitemia within BSG negative (KO) population is shown in red with BSG positive (rescue) population shown in blue **B)** Relative invasion efficiencies normalized to NT control. n = 3 Filled circles represent assay 1 with each data point an average of 3 technical replicates at a different starting parasitemia, open circles represent 2nd independent culture, open triangles represent reticulocytes from the same second culture using an independent parasite preparation on a different day.

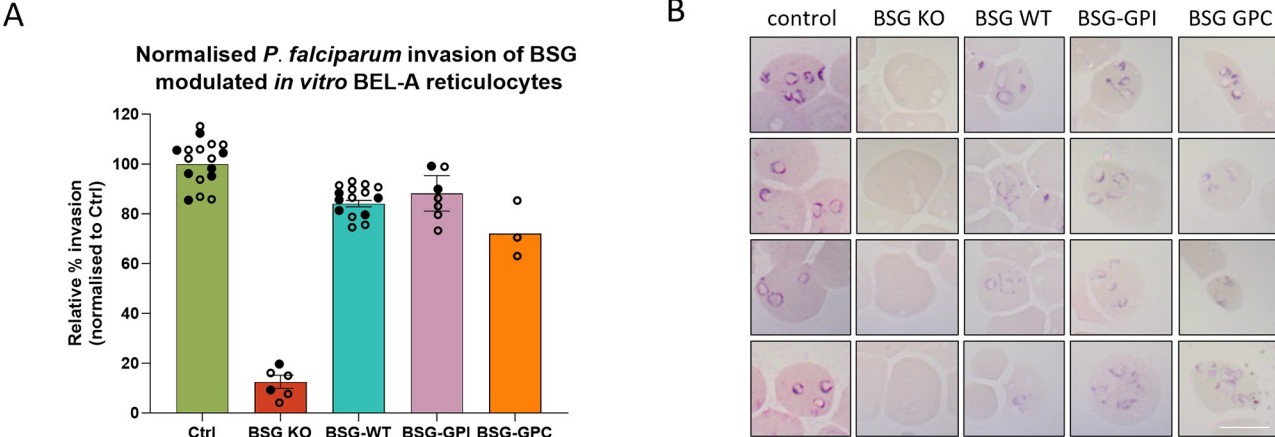

**Fig 4. Invasion of modified BEL-A derived reticulocytes confirms dispensability of basigin transmembrane domain and interaction with MCT1 for merozoite invasion. A)** Relative invasion efficiencies normalized to unedited BEL-A derived reticulocytes. n = 2 independent cultures, starting parasitemia 8%. Filled circles illustrate technical replicates from assay 1, open circles from assay 2. **B)** Illustrative cytospins provide visual confirmation of ring stage parasites in invaded samples.

observed in BSG KO cells rescued with BSG WT, BSG GPI or BSG GPC. Giemsa stained cytospin images shown in Fig 4B provide visual confirmation of rings in each rescue population, whilst no rings are observed in BSG KO reticulocytes. These data confirm the conclusion that the transmembrane domain of basigin that mediates interaction with MCT1 is dispensable for invasion.

## Discussion

Determination of the essentiality of the molecular interaction between merozoite PfRh5 and red blood cell basigin has proved to be a seminal discovery in malaria invasion research [3], spawning an array of cell biological, structural and therapeutic development studies centred around this crucial interaction [18–22]. Collective knowledge regarding the timing of this interaction within the coordinated invasion process [19,20], the existence of PfRh5 as part of a larger multiprotein complex of parasite proteins [23,24] and information regarding location of epitopes of invasion inhibitory antibodies [25,26] has burgeoned. However the mechanism by which PfRh5-basigin interaction facilitates downstream continuation of invasion remains a subject of controversy, whilst the evolutionary bases that presided to the specific targeting of basigin by the parasite for such an essential process and the downstream consequences within and beneath the host red blood cell membrane remain unclear.

Recently Jamwal et al. reported that recombinant PfRh5 binds to complexes of basigin-MCT1 and basigin-PMCA1/4 with greater affinity than basigin ectodomain alone and further demonstrated that growth inhibitory antibodies that do not disrupt binding of PfRh5 to recombinant basigin ectodomain do inhibit binding to basigin within either of these heteromeric complexes [8]. These data suggest that the interaction between MCT1 (or other proteins that interact laterally via the transmembrane helix) and basigin may be important for invasion within a cellular context.

Whilst our data are not contradictory to these findings, indeed it seems likely that basigin presented within its native complex would interact preferentially compared to in isolation, we show that neither this increased affinity for PfRh5 binding, nor the presence of lateral interactions that require the transmembrane domain of basigin are functionally required for its facilitation of *Plasmodium falciparum* invasion.

The precise bases and molecular consequences of PfRh5 binding to basigin are a subject of debate. Blockade of this interaction by antibodies to either PfRh5 or basigin causes an arrest of invasion immediately prior to the formation of the tight junction, also preventing the appearance of the transient pulse of calcium at the merozoite erythrocyte interface that triggers or accompanies this process. PfRh5 exists as part of a pentameric PCRCR complex with other parasite proteins CyRPA, PfRipr [23] and more recently PfPTRAMP and PfCSS [24] shown to interact. PfRh5 and PfRipr have been proposed to insert into the membrane following conformational changes upon binding to basigin, facilitating formation of a pore that enables invading merozoites to inject components into the erythrocyte cytoplasm [23]. However the viability of this mechanism has recently been challenged, with locking disulphide bonds introduced into PfRh5 that prevent such necessary conformational changes failing to inhibit invasion [27].

Consequences of basigin binding by the merozoite within the host cell are also a subject of tremendous interest, phosphorylation of red blood cell cytoskeletal components in response to merozoite engagement with host receptors is widely believed to play a contributory role in facilitating tight junction formation and or parasite internalisation [21,28–30]. Aniweh et al reported calcium influx and phosphorylation of cytoskeletal components induced by basigin binding of recombinant PfRh5 alone, emphasising the importance of basigin dimerization for calcium influx [21].

In addition to complexes with MCT1, basigin has also been reported to exists as hetero-meric complexes with PMCA4 [8], a plasma membrane calcium ATPase required for cellular efflux of calcium in red blood cells, with this interaction also mediated via the transmembrane domain of basigin. Single nucleotide polymorphisms in the ATP2B4 gene that encodes PMCA4 have been associated with resistance to severe malaria in several Genome Wide Association Studies [31–33] leading to widespread interest in the putative contribution of this protein in *P. falciparum invasion*, growth and pathogenesis. However, while the identification of a calcium ATPase as interacting partner of basigin (engagement of which by PfRh5 is accompanied by a transient pulse of calcium during merozoite invasion [19]) provided initial excitement, subsequent data do not support functional modulation of PMCA activity based upon PfRh5 interaction with basigin [8]. Whilst we did not explore effects of basigin TM domain removal on PMCA directly, absence of the effect on invasion of deletion of this interface, as in the case of MCT1, by extension excludes a requirement for direct interaction between basigin TM domain and PMCA proteins for successful *P. falciparum* invasion. The means by which basigin facilitates intracellular signalling that modifies the host cell remains unclear, with our previous work demonstrating the dispensability of the cytoplasmic domain of basigin [10] for invasion extended here to encompass the transmembrane domain, lateral interactions thereof and specific host membrane context.

The data we present here demonstrates the requirement solely for basigin ectodomain presentation within the host red blood cell membrane for its facilitation of successful invasion. These data challenge the hypothesis that the context and established TM mediated pre-existing host cell interactions of basigin are crucial to its role in invasion. One explanation for these data therefore is that basigin acts solely as a binding site and activation signal for parasite mediated processes such as rhoptry release that can occur irrespective of the local context of the PfRh5 engagement. However it should also be noted that irrespective of means and context of anchorage within the plasma membrane, the basigin ectodomain itself will still have the capacity and potential to interact with ectodomains of itself and other proteins on the host and merozoite surfaces. Interestingly, Yong et al recently reported the existence of a complex of basigin with CD44 and the beta adrenergic receptor [34] (although neither protein was detected in detergent extracts of erythrocytes from a different study [8]. They hypothesise that a pre-existing basigin associated complex containing CD44 and other host proteins undergoes increased protein assembly and recruitment that subsequently activates cAMP signalling via the beta adrenergic receptor [34]. One alternative host oriented hypothesis therefore is that host protein complexes assemble actively at the site of invasion (irrespective of pre-existing membrane context) around the basigin ectodomain- PfRh5 interaction in a process that is mediated or modulated by alterations to the extracellular domain alone upon engagement, or the result of proximal recruitment of host proteins that interface with the basigin extracellular domain by engagement of other host and parasite proteins by EBA and Rh proteins at this site. Identification of these recruited proteins and their role in invasion and merozoite internalisation will be important in future studies.

In summary, we present here data that establishes the dispensability of the transmembrane and cytoplasmic domains of basigin together with interactions of this protein mediated via these regions for mediation of merozoite invasion by *Plasmodium falciparum*. Interaction of basigin with MCT1 is not a pre-requisite for invasion, whilst reduced levels of MCT1 does not inhibit invasion of reticulocytes. Similarly invasion is able to proceed unimpeded by membrane relocalisation of basigin through substitution of its transmembrane helix or through alteration of membrane anchorage from transmembrane helix to GPI anchor, suggesting an intriguingly basal host membrane context independent role in facilitation of invasion. These data contribute to our evolving understanding of events, requirements and downstream

consequences of the PfRh5- basigin interaction that is essential for invasion. Future investigation of induced assembly of host-parasite protein complexes at the site of tight junction formation and ongoing research to define conformational rearrangements and the nature of induced signalling that arises in both host and parasite will inform broad and varied efforts to disrupt this crucial step in invasion.

## Supporting information

**S1 Fig. Surface expression of basigin in BEL-A derived reticulocytes. A)** Flow cytometry dot plots illustrating surface basigin expression of BEL-A derived reticulocytes as indicated **B)** Flow cytometry histograms illustrating comparative surface expression of basigin as assessed by HIM6 labelling on BEL-A derived reticulocyte populations.
(TIF)

**S2 Fig. CD44 surface presentation in BSG modified reticulocytes assessed with different CD44 antibodies.** Bar chart illustrating expression of CD44 as assessed with anti-CD44 BRIC222, KZ1 and BRIC235 on indicated BEL-A derived reticulocytes normalized to unedited cells. Data from at least 3 independently differentiated experiments are presented. A Kruskal-Wallis comparison followed by Dunn's multiple comparison correction was performed to test for differences between groups. $p < 0.05$ was considered statistically significant.
(TIF)

**S3 Fig. Surface expression of basigin in CD34[+] cell derived reticulocytes. A)** Flow cytometry dot plots illustrating basigin surface expression in BSG KO and rescue populations as assessed by HIM6 labelling **B)** Flow cytometry histograms illustrating basigin surface expression of successfully rescued populations by comparison to RBC, NT Ctrl reticulocytes and BSG KO.
(TIF)

**S1 Data. Numerical values used for generation of figures.**
(XLSX)

## Acknowledgments

The authors wish to thank Andrew Herman from the University of Bristol Faculty of Biomedical Sciences Flow Cytometry Facility for cell sorting support and Vincent Petit (Metafora Biosystems) for kind provision of HC2.RBD.rFC. MS and JT are grateful to Myriam Boyer of Montpellier Rio Imaging for support in cytometry experiments.

## Author Contributions

**Conceptualization:** Timothy J. Satchwell.

**Data curation:** Nadine R. King, Catarina Martins Freire, Timothy J. Satchwell.

**Formal analysis:** Nadine R. King, Timothy J. Satchwell.

**Funding acquisition:** Marc Sitbon, Ashley M. Toye, Timothy J. Satchwell.

**Investigation:** Nadine R. King, Jawida Touhami, Timothy J. Satchwell.

**Methodology:** Timothy J. Satchwell.

**Project administration:** Timothy J. Satchwell.

**Resources:** Jawida Touhami, Marc Sitbon.

**Supervision:** Marc Sitbon, Timothy J. Satchwell.

**Visualization:** Timothy J. Satchwell.

**Writing – original draft:** Timothy J. Satchwell.

**Writing – review & editing:** Nadine R. King, Catarina Martins Freire, Marc Sitbon, Ashley M. Toye, Timothy J. Satchwell.

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
