## [Decision Letter · Decision Letter 0]

30 Dec 2023

Dear Dr. Satchwell,

Thank you very much for submitting your manuscript "Basigin mediation of *Plasmodium falciparum* red blood cell invasion does not require its interaction with monocarboxylate transporter 1" for consideration at PLOS Pathogens. As with all papers reviewed by the journal, your manuscript was reviewed by members of the editorial board and by several independent reviewers. The reviewers appreciated the attention to an important topic. Based on the reviews, we are likely to accept this manuscript for publication, providing that you modify the manuscript according to the review recommendations.

You will see from the reviewers' comments that they found the manuscript interesting, important and convincing. They had only relatively minor comments/suggestions for consideration. Please read through the comments and make appropriate modifications to the paper. Additional experiments are not required, although you might decide to perform some relatively minor additional work (e.g. giemsa images, Western blot, etc).

Sincerely,

Kirk W. Deitsch

Academic Editor

PLOS Pathogens

Margaret Phillips

Section Editor

PLOS Pathogens

Kasturi Haldar

Editor-in-Chief

PLOS Pathogens

orcid.org/0000-0001-5065-158X

Michael Malim

Editor-in-Chief

PLOS Pathogens

orcid.org/0000-0002-7699-2064

Reviewer Comments (if any, and for reference):

Reviewer's Responses to Questions

**Part I - Summary**

Reviewer #1: This paper presents an interesting and compact study in which the authors employ their expertise in genetic manipulation of primary haematopoietic stem cells, which can be invaded by the malaria parasites, Plasmodium falciparum, to assess the importance of the transmembrane domain of the receptor basigin.

This is an interesting question as basigin is required for the invasion of erythrocytes by this parasite, and a variety of theories exist about how this happens. These include signalling through basigin, which the authors have already tackled by replacing basigin with a version lacking its cytoplasmic domains, in a previous publication. Basigin also forms membrane complexes with the membrane proteins PMCA and MCT1, through their transmembrane helix. The authors here replace the transmembrane helix of basigin with that of a different protein or with a GPI anchor and assess the effect on invasion.

It appears as though this change has blocked the binding of basigin to MCT1 (as would be expected based on the structure) due to the change in MCT1 surface expression which occurs as a result.

They then proceed to show that versions of basigin which only have the basigin ectodomain with a different TM or GPI can complement WT basigin, forming reticulocytes which can be invaded by parasites. This is a novel experiment and an interesting outcome as it throws into question all of the models in which basigin acts through transmembrane interactions with other proteins and/or signalling into the erythrocyte.

In summary, this is a highly focused paper, with a single key experiment, repeated with two different knockout lines in a robust manner. It gives a clear conclusion, with important consequences for a major malaria vaccine immunogen.

Reviewer #2: Plasmodium falciparum parasites use their RH5 protein to bind to the RBC receptor basigin (BSG) to facilitate invasion of human RBCs. RH5 is attached to the surface of the invasive merozoite parasite stage via a chain of 4 other proteins collectively known as the PCRCR complex. BSG is anchored to the RBC surface via a transmembrane domain (TMD) and binds to either of two other RBC proteins called MCT1 and PMCA1/4. During invasion a calcium ion flux forms at the synapse between the merozoite and RBC leading to the proposal that there may be some rearrangement of the RH5/BSG complex which involves BSG’s transmembrane domain and results in a pore in the RBC membrane. BSG’s interaction with MCT1 facilitated by BSG’s TMD, might also have an important role in invasion. To explore this further, the authors modify erythroblasts to knockout BSG or replace it with chimeric versions of BSG in which its membrane anchor has been replaced with that of another protein or with a GPI anchor. As anticipated, BSG knockout reticulocytes could not be invaded but retics with the alternative membrane anchored forms of BSG can be invaded despite not binding MCT1. Overall, the paper shows that only the ectodomain of BSG is needed for successful parasite invasion of RBCs.

Reviewer #3: In this study the authors are investigating molecular details of basigin-MCT1 interaction in the context of Plasmodium falciparum invasion of human erythrocytes.

Erythrocyte invasion is a crucial and mandatory step for the proliferation of the malaria parasite. The multi-step process involves a complex and still only partially understood molecular machinery. It is mediated by receptor-ligand interaction that allows this cell-cell interactions. In P. falciparum it comprises parasite ligands such as the P. falciparum erythrocyte binding-like (PfEBL) and reticulocyte binding protein homologue (PfRh) protein families as well as erythrocyte surface structure such as glycophorin A or basigin. Individual disruption of the PfRh- and PfEBL-encoding genes does not result in an erythrocyte invasion phenotype exposing functional redundancy within these receptor–ligand interactions - with one exception: PfRH5 is the only member of either family showns to be necessary for erythrocyte invasion in all tested parasite strains. PfRh5 forms a complex with the parasite Ripr (Rh5 interacting protein) and CyRPA (cysteine-rich protective antigen) to interact with the erythrocyte membrane protein basigin. Basigin itself is part of a multiprotein complexes that includes the monocarboxylate transporter MCT1.

Following up on some of their previous work the authors present an elegant study that feeds on genetically manipulated in vitro derived reticulocytes expressing different Basign variants. The authors demonstrate that merozoite invasion of reticulocytes is unaffected by disruption of basigin MCT1 interaction via various mutations. They conclude, that the MCT1-Basigin interaction is not a requisite for parasite invasion. Overall, the experimental set-up is elegant and well designed and the results are presented clearly. The derived some new insights that challenge some previous assumptions of the complex molecular interplay involved in P. falciparum erythrocytes invasion.

**Part II – Major Issues: Key Experiments Required for Acceptance**

Reviewer #1: (No Response)

Reviewer #2: None

Reviewer #3: I have only two major points to discuss:

1. Fig. 1A/B: Quantification of basign and and MCT1 by FACS in mutant cell lines. As a second line of evidence IFA based quantitative microscopy would be nice to probe into the effect of the absence of basigin on reticulocyte surface expression of MCT1 and vice versa. Additional Western Blot or mass spectrometry-based expression analysis could also contribute to a more comprehensive description of the generated mutant cell lines and validate the given FACS-based analysis and the main conclusions.

Out of curiosity: Did the authors contemplated on a rescue of the MCT1 KO line on an over expressing BSG parental cell line? Feasible?

2. Figure 3+4: Invasion assays.

2.1 What was the rationale for using up to 8% schizonts as starting parasitemia? If egress is normal within their experimental set up it translate to a massive amount of free, invasion capable merozoites although Figure 3A shows a maximum of 8% parasitemia as an end point. Is the invasion capability of 3D7 parasites of the engineered cells that low? Why? What are the implications? Is it also parasite strain dependent?

2.2 Some additional documentation of the invasion assays by giemsa stained smears at multiple time points (e.g. schizont, post egress, end point) showing for instance normal and complete egress of the used schizont material, the presence of rings (and a substantial amount of non-invaded merozoites) after reinvasion would give the reader another possibility to assess this invasion assay.

2.3 As shown in Figure 4B, there are multiple infections with up to five parasites. Is this agin due to the “reticulocyte nature” of the host cells? In comparable invasion assays using “normal erythrocytes” multiple infection happen but are rather the exception than the rule.

2.4 Arising from this: Does the used flow cytometry approach i) is able to differentiate between single and multiple infections, ii) are cells with multiple infections quantified as one invasion event? and iii) is the number of multiple infections comparable between the different conditions/cell lines? This appears relevant if the phenotypic consequences of the different variants are assessed.

**Part III – Minor Issues: Editorial and Data Presentation Modifications**

Reviewer #1: First I think that the title and selling of the paper around the interaction with MCT1 doesn't quite fit the data. The main conclusion is broader than being about MCT1, but instead shows that any function of the TM of basigin is not required for invasion. Basigin also binds to PMCAs in erythrocytes, through its TM domains and it is equally interesting that this interaction is not required for invasion. I did consider asking the authors to assess the effect of their mutant basigins on PMCA surface expression, but decided against, as it is likely that the TM replacements will also disrupt PMCA interactions. However, it would be good to broaden the focus of the paper to include discussion of PMCA and signalling too. In fact I would change the title to broaden it from MCT1 to all the other effects of TM changes.

The major omission was the FACS data showing that the different basigin variants are expressed properly and surface exposed. Figure 2B could do with better labelling and would be valuable to see equivalent data in SI for each of the mutants. I presume that there is no issue here as the basigin variants were invaded by parasites, but it is important to present these validations of correct surface exposure of the variant basigins for both cell types.

Reviewer #2: 1. Overall this is a very well-prepared manuscript with no major faults. I would like to know however, if the retics invaded with the chimeric versions of BSG complete their cell cycles and continue to proliferate. This could be done by seeding the newly invaded retics into a dish of normocytes to see if they proliferate.

2. For reference 1, use the latest WHO World Malaria Report.

3. Line 61. Should ‘bases’ be ‘basis’?

4. Please clarify what is meant on line 350 ‘…may be important for invasion within a cellular context.’

Reviewer #3: 1. Most of the figures need some adjustments and harmonization. Font and font sizes are varying within and between figures.

2. Figure 1D/E – What does the black curve represent?

3. Figure 3B: y-axis has no legend.

4. Figure 4 A/B: What was the starting parasitemia for the shown invasion assay?

5. The authors are refering to the WHO Malaria report 2021. They could update the numbers with a newer report.

6. Again out of the reviewer curiosity (but it might be also interesting for the general reader): Why did the authors focus their study on MCT1 and not the ATPase PMCA1/4 within the Basign protein complex?

PLOS authors have the option to publish the peer review history of their article (what does this mean?). If published, this will include your full peer review and any attached files.

Reviewer #1: No

Reviewer #2: No

Reviewer #3: No

Figure Files:

Data Requirements:

Reproducibility:

References:

---

## [Editor Report · Decision Letter 1]

19 Jan 2024

Dear Dr. Satchwell,

We are pleased to inform you that your manuscript 'Basigin mediation of * Plasmodium falciparum * red blood cell invasion does not require its transmembrane domain or interaction with monocarboxylate transporter 1' has been provisionally accepted for publication in PLOS Pathogens.

Best regards,

Kirk W. Deitsch

Academic Editor

PLOS Pathogens

Margaret Phillips

Section Editor

PLOS Pathogens

Michael Malim

Editor-in-Chief

PLOS Pathogens

orcid.org/0000-0002-7699-2064
---

## [Editor Report · Acceptance letter]

30 Jan 2024

Dear Dr. Satchwell,

We are delighted to inform you that your manuscript, "Basigin mediation of * Plasmodium falciparum * red blood cell invasion does not require its transmembrane domain or interaction with monocarboxylate transporter 1," has been formally accepted for publication in PLOS Pathogens.

Best regards,

Michael Malim

Editor-in-Chief

PLOS Pathogens

orcid.org/0000-0002-7699-2064